# PAIRWISE ADVERSARIAL TRAINING FOR UNSUPERVISED CLASS-IMBALANCED DOMAIN ADAPTATION

## ABSTRACT

Unsupervised domain adaptation (UDA) has become an appealing approach for knowledge transfer from a labeled source domain to an unlabeled target domain. However, when the classes in source and target domains are imbalanced, most existing UDA methods experience significant performance drop, as the decision boundary usually favors the majority classes. Some recent class-imbalanced domain adaptation (CDA) methods aim to tackle the challenge of biased label distribution by exploiting pseudo-labeled target data during training process. However, these methods still suffer from the issues with unreliable pseudo labels and error accumulation during training. In this paper, we propose a pairwise adversarial training approach for class-imbalanced domain adaptation. Unlike conventional adversarial training in which the adversarial samples are obtained from the $\ell_p$ ball of the original data, we generate adversarial samples from the interpolated line of the aligned pairwise samples from source domain and target domain. Experimental results and ablation studies show that our method achieves considerable improvements on benchmarks compared with the state-of-art CDA methods.

'

## 1 INTRODUCTION

Unsupervised domain adaptation (UDA) aims to achieve knowledge transfer from a labeled source domain to an unlabelled target domain. Recent years have witnessed the significant progress of UDA based on deep neural networks (Pei et al., 2018; Cui et al., 2020; Hu et al., 2020; Liang et al., 2020; Na et al., 2021). Most of existing UDA methods assume that only covariate shift occurs in the source domain and target domain, while the label distributions in two domains are identical. However, this assumption may not hold in real-world applications. For instance, in wild-life pictures, the commonly seen animals such as rabbit and deer appear more frequently than the rare animal such as panda and crocodile. Public datasets such as DomainNet (Peng et al., 2019) and and MSCOCO (Lin et al., 2014) exhibit imbalanced class distribution. Figure 1 illustrates the imbalanced label distributions in the Real domain and Sketch domain from the DomainNet dataset.

To address the issue of imbalanced label distributions in domain adaptation, some recent studies (Wu et al., 2019; Tan et al., 2020; Jiang et al., 2020) try to jointly model the conditional feature distribution shift and label distribution shift (LDS). This problem is referred to as Class-imbalanced Domain Adaptation (CDA). Let $x$ and $y$ denote the samples and labels, respectively. $p$ and $q$ separately represent the probability distribution of source domain and target domain. The common assumptions in UDA involve the covariate shift (i.e., $p(x) \neq q(x)$) and identical label distribution (i.e., $p(y) = q(y)$). In CDA, however, apart from the covariate shift, both the conditional feature shift and label shift exist, i.e., $p(x|y) \neq q(x|y), p(y) \neq q(y)$.

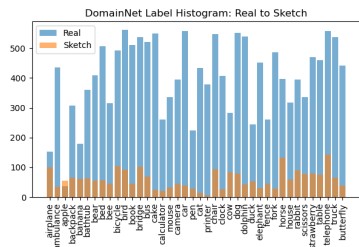

Figure 1: Illustration of label distribution shift in DomainNet dataset.

CDA is a more challenging task than UDA. Recent studies (Tan et al., 2020) have demonstrated that the mainstream UDA methods will suffer significant performance drop, as the classifier will favor

the majority classes. Only a few CDA approaches have been proposed by far. In Tan et al. (2020)'s work, the negative effect of label shift is reduced by exploiting the pseudo labelled target samples via self-training. Jiang et al. (2020) use an implicit sampling method based on pseudo labels to align the joint distribution between features and labels. However, one critical problem of these methods is that the pseudo labels are likely to suffer from ill-calibrated probabilities (Guo et al., 2017), and thus the unreliable pseudo labels will cause error accumulation during the training process, which will largely degrade the model performance.

Augmenting training data has been proven as an effective strategy to tackle the issue of biased label distributions in class-imbalance learning (Chawla et al., 2002; Chou et al., 2020). In addition to the traditional data augmentation techniques, adversarial training is also capable of generating semantically meaningful synthetic samples that help enhance the robustness of models. However, these approaches only consider a single domain, and they cannot be directly applied to solve the CDA problem. In this paper, we propose a pairwise adversarial training (PAT) approach that augments training data for class-imbalanced domain adaptation. Unlike conventional adversarial training in which the adversarial samples are obtained from the $\ell_p$ ball of the original data, we obtain the semantic adversarial samples from the interpolated line of the aligned pair-wise samples from source domain and target domain. Moreover, a class-imbalanced semantic centroid alignment strategy is designed to explicitly align the source and target domains in the feature space.

The main contributions of this paper are three-fold. (1) We propose a novel pairwise adversarial training approach that generates adversarial samples from pairs of samples across the source and target domains, and further exploits these samples to augment training data. (2) We propose a new optimization algorithm to solve pairwise adversarial training problem. (3) We conduct extensive evaluations on benchmark datasets, and results show that our approach obtains competing performance compared with state-of-art CDA methods.

## 2 RELATED WORK

In this section, we briefly introduce three relevant research topics, including unsupervised domain adaptation, class-imbalanced domain adaptation and adversarial training.

**Unsupervised Domain Adaptation**. In recent years, unsupervised domain adaption (UDA) has attracted increasing attention. Existing UDA methods could be roughly categorized into two groups, including the discrepancy-based methods and adversarial-based methods. The discrepancy-based methods usually align source and target feature distributions in the embedding space using various statistical distance metrics, such as Maximum Mean Discrepancy (MMD) (Long et al., 2016; 2017; Kang et al., 2019), Correlation Alignment (CORAL) (Sun & Saenko, 2016), and Wasserstein distance (Lee & Raginsky, 2018; Shen et al., 2018; Balaji et al., 2019). On the other hand, the adversarial-based methods focus on learning domain invariant features via domain adversarial training (Ganin et al., 2016; Shu et al., 2018; Pei et al., 2018; Saito et al., 2018; Deng et al., 2019; Yu et al., 2019). Recently, Zhang et al. (2019) proposed the margin disparity discrepancy (MDD) to measure the discrepancy of two domains with generalization bounds. This theory is tailored into an adversarial learning algorithm for domain adaptation. Unlike other adversarial learning based UDA methods that align two domains by confusing a domain discriminator, MDD aligns two domains by minimizing the maximum margin disparity discrepancy of an optimal classifier $f$ and an auxiliary classifier $f'$. The optimization problem of MDD is formulated as:

$$\min_{f,\psi} \varepsilon(\mathcal{D}_s) + \eta \mathcal{D}_\gamma(\mathcal{D}_s, \mathcal{D}_t), \tag{1}$$

$$\max_{f'} \mathcal{D}_\gamma(\mathcal{D}_s, \mathcal{D}_t), \tag{2}$$

where $\varepsilon$ is the classification loss on the source domain and $\mathcal{D}_\gamma$ measures the discrepancy of source domain and target domain. Specifically,

$$\varepsilon(\mathcal{D}_s) = \mathbb{E}_{(x^s, y^s) \sim \mathcal{D}_s} \mathcal{L}(f(\psi(x^s)), y^s), \tag{3}$$

$$\mathcal{L}_{adv} = \mathcal{D}_\gamma(\mathcal{D}_s, \mathcal{D}_t) = \mathbb{E}_{x^t \sim \mathcal{D}_t} \mathcal{L}'(f'(\psi(x^t)), f(\psi(x^t))) - \gamma \mathbb{E}_{x^s \sim \mathcal{D}_s} \mathcal{L}(f'(\psi(x^s))), f(\psi(x^s)), \tag{4}$$

$\mathcal{L}$ is cross-entropy function, and $\mathcal{L}'(f'(\psi(x^t)), f(\psi(x^t))) = \log(1 - \sigma_{y'}(f'(\psi(x^t))))$. $y'$ is the pseudo label generated from an optimal classifier. MDD is the backbone of our method.

**Class-imbalanced Domain Adaptation**. As a branch of domain adaptation, the class-imbalanced domain adaptation (CDA) aims to deal with data with biased class distribution. Tan et al. (2020) might be the first one to investigate the CDA problem, and they exploited the pseudo labelled target data to reduce the negative effect of label shift. Wu et al. (2019) proposed the asymmetrically-relaxed distances as replacement of the standard ones under biased label distribution. Jiang et al. (2020) adopted the implicit sampling strategy to ensure class alignment at the minibatch level. Prabhu et al. (2021) avoided the use of highly unreliable pseudo labels by assessing the reliability of target data with predictive consistency under random image transformations. Our method refrains from the exploitation of pseudo labeled target data directly in the training process, while reducing the effect of biased label shift by incorporating the semantic adversarial samples into the training process.

**Adversarial Training**. Adversarial training (AT) (Szegedy et al., 2014; Goodfellow et al., 2015) is an effective regularization method for enhancing the robustness and generalization ability of deep learning models. In particular, adversarial samples are incorporated in the model training process, which are intentionally designed to deceive the deep learning model by adding small perturbation on the original data. Furthermore, virtual adversarial training (VAT) has been proposed (Miyato et al., 2018), which seeks the adversarial direction for regularization without using label information. Both AT and VAT have been employed to tackle the standard UDA problems Shu et al. (2018). However, to the best of our knowledge, our work is the first attempt to address the class-imbalanced domain adaptation problem using adversarial training.

## 3 PROPOSED APPROACH

In this section, we first give the problem definition of CDA, and then present the details of the proposed pairwise adversarial training approach. Finally, we introduce how to integrate the pairwise adversarial training with MDD to address the CDA problem.

### 3.1 PROBLEM DEFINITION

In class-imbalanced domain adaptation, both the source and target domains suffer from label distribution shift. We are given a source domain $\mathcal{D}_s = \{(x_i^s, y_i^s)\}_{i=1}^{N_s}$ with $N^s$ labelled samples and a target domain $\mathcal{D}_t = \{x_i^t\}_{i=1}^{N_t}$ with $N^t$ unlabelled samples. Each domain contains $K$ classes, and the class label is denoted as $y^s \in \{0, 1, 2, ..., K-1\}$. Let $p$ and $q$ denote the probability distributions of the source domain and target domain, respectively. We assume that both the covariate shift (i.e., $p(x) \neq q(x)$) and label distribution shift (i.e., $p(y) \neq q(y)$ and $p(x|y) \neq q(x|y)$) exist in two domains. Our goal is to train a model that can learn domain invariant features, reduce the gap between source and target domains, and mitigate the label distribution shift. The model typically consists of a feature extractor $\psi : \mathcal{X} \to \mathcal{Z}$ and a classifier $f : \mathcal{Z} \to \mathcal{Y}$ that aims to minimize the target risk.

### 3.2 PAIRWISE ADVERSARIAL TRAINING (PAT)

We investigate how to mitigate the challenging issue of label distribution shift in CDA, as illustrated in Figure 1. Previous studies (Tan et al., 2020) found that when the source domain is imbalanced, the model performance on target domains will be significantly dropped, especially when the target domain is also imbalanced. An intuitive solution is to augment the training data in two domains, such that the model training would not be dominated by the majority classes in either domain. However, this task is not trivial, considering the mixed effects of domain gap and imbalanced class distributions. Inspired by adversarial training, we aim to create adversarial samples to augment training data. In adversarial training, the adversarial samples will be exploited to enhance the robustness and generalization ability of model. The loss function of adversarial training is:

$$\mathcal{L}_{ce}(x + \delta^*, y; \theta)$$
$$\text{where} \quad \delta^* := \arg\max_{||\delta||_p \leq \epsilon} \mathcal{L}_{ce}(x + \delta, y; \theta), \tag{5}$$

where $x$ is the original sample, $y$ is the ground-truth label of $x$, $\theta$ refers to model parameters, and $\delta$ is the perturbation added to $x$.

The existing adversarial training methods could not be directly used to tackle the CDA problem for two reasons. First, existing methods simply generate adversarial samples within the neighborhood

of the original samples, but they could not mitigate the gap between source and target domains. Second, existing methods treat majority classes and minority classes equally, so they are unable to address the class imbalance issue. In this paper, we propose pairwise adversarial training that generates adversarial samples from the linear interpolation of source and target samples and meanwhile reduces the domain discrepancy. In the following, we will introduce two key components of PAT, including the generation of interpolated adversarial samples and semantic centroid alignment.

### 3.2.1 INTERPOLATED ADVERSARIAL SAMPLE GENERATION

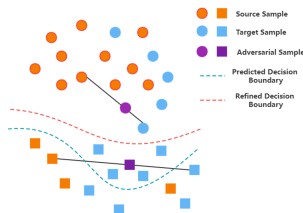

As shown in Figure 2, we will generate adversarial samples on the interpolated line from a source sample to a target sample of the same class. The interpolated adversarial samples (IAS) should have the same semantic meaning as their corresponding source/target sample pair, although they are not perfectly aligned with the original samples. We explicitly address the data imbalance issue in the source domain by exploiting interpolated adversarial samples and aligning the source and target domains. As a result, the generalization ability of the unbiased model will be improved and the data imbalance issue in the target domain could be implicitly addressed. For the $k$th class, the interpolated adversarial samples can be defined as:

Figure 2: Illustration of an interpolated adversarial sample, which lies on the interpolated line of the source and target samples.

$$\mathcal{X}_k^{adv} = \{x_k^{adv} | x_i^{adv} = x_i^s + \lambda(x_i^t - x_i^s), \lambda \in [0,1)^C, y_i^s = \hat{y}_i^t = k\}, \quad (6)$$

where $x_i \in \mathbb{R}^{C \times H \times W}$, $\lambda$ is the coefficient measuring the contributing weights of the source sample and target sample. $\hat{y}_i^t$ is the pseudo label of the target sample which is generated from the backbone domain adaptation model (e.g., the optimal classifier $f$ in MDD). It is used for match of the corresponding source sample. In our work, though the pseudo labels might be incorrect, our PAT method will not suffer from the potential error accumulation issue. First, we choose the target samples with high confidence. Second, even if the pseudo label is incorrect, there is still a chance that the interpolated adversarial sample will be generated within the boundary as expected. Specifically, the misclassified target samples often appear near the decision boundary. So, even if the target sample is from a different class, the adversarial sample generated from the pair of source sample and target sample may not violate the decision boundary. Finally, the adversarial samples in our approach are generated dynamically, and the adverse effect of bad adversarial samples could be mitigated.

Then, the generation of interpolated adversarial samples could be achieved by solving the following optimization problem:

$$\mathcal{L}_{IAS} := D(\hat{x}^{adv}, y; \theta)$$
$$\text{where} \quad \hat{x}^{adv} = \underset{x^{adv} \in \mathcal{X}^{adv}}{\arg\max} \, D'(x^{adv}, y; \theta). \quad (7)$$

In Eq. (7), the outer minimization problem involves a standard cross-entropy loss function, i.e.,

$$D(\hat{x}^{adv}, y; \theta) = \mathcal{L}_{CE}(\hat{x}^{adv}, y; \theta) = -\log(\sigma_y(f(\psi(\hat{x}^{adv})))), \quad (8)$$

where $\psi : \mathcal{X} \to \mathcal{Z}$ denotes a feature extractor, $f : \mathcal{Z} \to \mathcal{Y}$ denotes a classifier, and $\sigma$ is the softmax function. For the inner maximization problem, we use a modified cross-entropy function proposed by Goodfellow et al. (2014). The modified loss function can alleviate the problem of gradient exploding or vanishing when the entropy loss is maximized. The loss function of the inner maximization problem is written as:

$$D'(x^{adv}, y; \theta) = \mathcal{L}'_{CE}(x^{adv}, y; \theta) = \log(1 - \sigma_y(f(\psi(x^{adv})))). \quad (9)$$

Several optimization algorithms, such as the fast gradient sign method (Goodfellow et al., 2015) and projected gradient descent (Madry et al., 2018), have been commonly used for adversarial training. However, these algorithms aim to obtain the adversarial samples in the $\ell_\infty$ ball of the original samples, which cannot be directly applied to solving our problem. In our case, the interpolated adversarial samples are confined on the interpolated line of source data and target data. We propose a new optimization algorithm to solve the inner maximization optimization in Eq. (7). We initialize the interpolated adversarial samples with random $\lambda$ and update them by back propagation in each iteration. The main procedures of our algorithm are summarized in Algorithm 1.

---

**Algorithm 1** Solving the maximization problem in PAT

---

**Input:** Source samples from a mini-batch $\{(x_i^s, y_i^s)\}_{i=1}^{B_s}$, target samples and their pseudo labels $\{(x_i^t, \hat{y}_i^t)\}_{i=1}^{N_t}$, probability threshold of each class $\{P_k\}_{k=1}^K$

**Output:** Interpolated adversarial samples $\{\hat{x}_i^{adv}\}_{i=1}^{N_{adv}}$
  **for** each source sample $x_i^s$ in the mini-batch **do**
    **if** rand() $> P_k(k = y_i^s)$ **then**
      Choose one target sample $x_i^t$ with pseudo label equals to $k$
      Initialize $x^{adv}$ with random $\lambda$ using Eq. (6)
      **repeat**
        Calculates the gradient of $\lambda$ with loss function in Eq. (7). $g_\lambda = \alpha \nabla_\lambda D'(x^{adv}, y; \theta)$
        Update $\lambda$ with the gradient, $\lambda \leftarrow \lambda + g_\lambda$
        Clip the $\lambda$ between 0 and 1, $\lambda \leftarrow \lambda \cdot \text{clip}(0, 1)$
        Update adversarial sample $\boldsymbol{x}^{adv}$ with new $\lambda$ using Eq. (6).
      **until** the optimization converges
    **end if**
  **end for**

---

Generating interpolated samples has been explored in literature, such as mix-up (Chou et al., 2020) and its variants. However, mix-up based methods and our method have significant differences. First, mix-up based methods focus on the single-domain classification or standard domain-adaptation problems, while our method focuses on the data imbalance problem in domain adaptation. Second, the ideas for generating samples and labels in mix-up and our method are different. Mix-up creates virtual samples and their labels from two randomly chosen samples. While in our method, an adversarial sample is generated using a pair of samples from two domains with the same label. Third, mix-up needs to manually define a hyper-parameter to control the strength of interpolation. However, in our method, the parameter for controlling the strength of interpolation is adaptively updated along with adversarial training.

### 3.2.2 CLASS-IMBALANCED SEMANTIC CENTROID ALIGNMENT

Without careful control of the generation mechanism, the interpolated adversarial samples may not alleviate the issue of imbalanced class distribution. Moreover, although the interpolated adversarial samples bridge the source and target domains to some extent, the discrepancy between source and target domains is not explicitly reduced. To address these issues, we propose to explicitly align the source and target domains with imbalanced class distributions using two strategies.

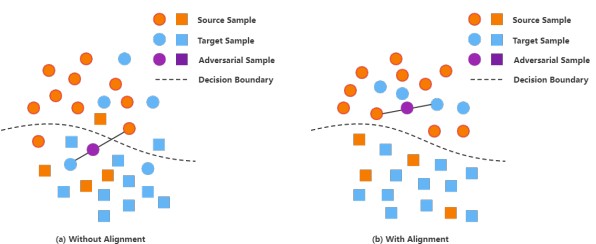

(a) Without Alignment      (b) With Alignment

Figure 3: Illustration of generation of adversarial samples (a) without centroid alignment and (b) with centroid alignment. In (a), adversarial samples have a larger chance to violate the decision boundary.

First, we propose a strategy to guide the generation of interpolated adversarial samples. For training samples in each mini-batch from the source domain, they should not have the equal opportunity to generate interpolated adversarial samples. Since the decision boundary usually favors the majority classes, the probability of generating adversarial samples for minority classes should be larger than that for majority classes. For the $k$-th class, we set a probability threshold $P_k$ as follows:

$$P_k = \frac{n_k}{n_{max} + \epsilon}, \tag{10}$$

where $n_k$ is the number of the samples from the $k$th class. $n_{max} = \max_k \{n_k\}_{k=1}^K$, and $\epsilon$ is the bias. For a specific class, if a random number $r \in [0, 1)$ is larger than the corresponding threshold, the adversarial samples will be generated, as shown in Algorithm 1. We also adopt class-balanced sampling on the source data to alleviate the biased occurrence of the majority classes. Specifically, each class will be selected with an equal chance, in order to reduce the model prediction bias towards the majority classes.

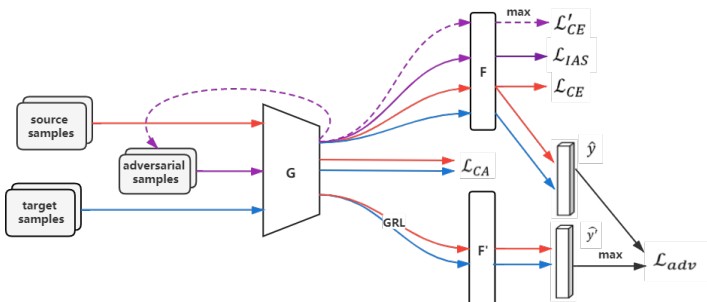

Figure 4: Illustration of our framework based on PAT and MDD. It includes a feature extractor $G$, an optimal classifier $F$, and an auxiliary classifier $F'$. The dashed line represents the data flow in pairwise adversarial training. The margin disparity discrepancy of two domains is diminished by aligning the two one-hot labels. We also explicitly align the pair of class conditioned samples by minimizing the distance of the centroids denoted as $\mathcal{L}_{CA}$.

Second, we incorporate the moving average centroid alignment (Xie et al., 2018) to align the conditional feature distributions of source and target domains by explicitly matching the centroids of two domains. As illustrated in Figure 3, without centroid alignment, the adversarial samples may be generated from a pair of samples in which one of the samples is misaligned to other classes, thus making the embedding of adversarial samples fall out of the decision boundary. With centroid alignment, we can eliminate the occurrence of such out-of-bound adversarial samples, and the interpolated adversarial samples could provide meaningful support for the minority class in the target domain. The loss function of moving average centroid alignment is defined as $\mathcal{L}_{CA} = \sum_{k=1}^{K} \text{dist}(\mathcal{C}_k^s, \mathcal{C}_k^t)$, where $\mathcal{C}_k^s$ and $\mathcal{C}_k^t$ denote the centroids of the $k$th class in the source domain and target domain, respectively. $\text{dist}()$ can be implemented by the Euclidean distance or cosine distance.

### 3.3 PAT FOR CLASS IMBALANCED DOMAIN ADAPTATION

The proposed PAT approach could be integrated with many existing domain adaptation frameworks to enhance their performance on class-imbalanced data. In this paper, we adopt MDD (Zhang et al., 2019) as an example backbone model and showcase how to integrated PAT with it.

The MDD framework consists of a feature extractor $G$, an optimal classifier $F$, and an auxiliary classifier $F'$. The loss function of MDD is introduced in Eq. (1) and Eq. (2). Adversarial samples designed to maximize the modified cross-entropy function $\mathcal{L}'_{CE}$ are generated from the pair of source data and target data. Finally, the overall loss function of our framework is:

$$\mathcal{L} = \mathcal{L}_{MDD} + \alpha \mathcal{L}_{IAS} + \beta \mathcal{L}_{CA}, \tag{11}$$

where $\alpha$ and $\beta$ are two trade-off parameters.

## 4 EXPERIMENTS

### 4.1 EXPERIMENTAL SETTINGS

**Office-31** is a widely used benchmark image dataset for domain adaptation (Saenko et al., 2010). It contains 31 classes in three domains: Amazon (**A**), Dslr (**D**) and Webcam (**W**). The standard Office-31 doesn't exhibit obvious label distribution shift (LDS), so a new imbalanced Office-31 is created by sampling from standard ones as suggested by Tan et al. (2020). The distribution conforms to Paredo distribution (Reed, 2001) and follows the **R**eversely-unbalanced **S**ource and **U**nbalanced **T**arget (**RS-UT**) protocol. Both the source domain and target domain have shifted label distribu-

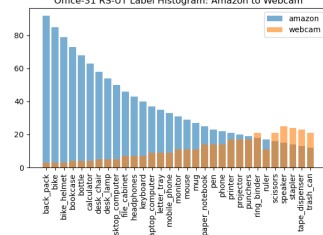

Figure 5: Biased label distribution shift on Amazon→Webcam from imbalanced Office-31.

tions, and the label distribution of source domain is a reversed version of that of target domain. **Office-Home** is a large benchmark dataset containing 65 classes of objects commonly found in office and home scenarios (Venkateswara et al., 2017). It has four domains: Real-World (**Rw**), Clipart

Table 1: Per-class average accuracy on Office-Home dataset with RS→UT label shift. Bold and underscore denote the best and second-best performing methods, respectively.

| Method | Rw→Pr | Rw→Cl | Pr→Rw | Pr→Cl | Cl→Rw | Cl→Pr | AVG |
|---|---|---|---|---|---|---|---|
| source [†] | 70.74 | 44.24 | 67.33 | 38.68 | 53.51 | 51.85 | 54.39 |
| BSP (Chen et al., 2019) [†] | 72.80 | 23.82 | 66.19 | 20.05 | 32.59 | 30.36 | 40.97 |
| PADA (Cao et al., 2018) [†] | 60.77 | 32.28 | 57.09 | 26.76 | 40.71 | 38.34 | 42.66 |
| BBSE (Lipton et al., 2018) [†] | 61.10 | 33.27 | 62.66 | 31.15 | 39.70 | 38.08 | 44.33 |
| MCD (Long et al., 2018) [†] | 66.03 | 33.17 | 62.95 | 29.99 | 44.47 | 39.01 | 45.94 |
| DAN (Long et al., 2015) | 69.35 | 40.84 | 66.93 | 34.66 | 53.55 | 52.09 | 52.90 |
| F-DANN (Wu et al., 2019) [†] | 58.56 | 40.57 | 67.32 | 37.33 | 55.84 | 53.67 | 53.88 |
| JAN (Long et al., 2017) [†] | 67.20 | 43.60 | 68.87 | 39.21 | 57.98 | 48.57 | 54.24 |
| DANN (Ganin & Lempitsky, 2015) [†] | 71.62 | 46.51 | 68.40 | 38.07 | 58.83 | 58.05 | 56.91 |
| MDD (Zhang et al., 2019) [†] | 71.21 | 44.78 | 69.31 | 42.56 | 52.10 | 52.70 | 55.44 |
| COAL (Tan et al., 2020) [†] | 73.65 | 42.58 | 73.26 | 40.61 | 59.22 | 57.33 | 58.40 |
| InstaPBM (Li et al., 2020) [†] | 75.56 | 42.93 | 70.30 | 39.32 | 61,87 | 63.40 | 58.90 |
| MDD+implicit (Jiang et al., 2020) [†] | 76.08 | 50.04 | 74.21 | 45.38 | 61.15 | 63.15 | 61.67 |
| SENTRY (Prabhu et al., 2021) [†] | 76.12 | **56.80** | 73.60 | **54.75** | 65.94 | 64.29 | 65.25 |
| Ours | **79.88** | 54.68 | **77.32** | 50.21 | **67.29** | **67.04** | **66.07** |

[†] Data of the baseline methods are cited from Prabhu et al. (2021) .

Table 2: Per-class average accuracy on DomainNet dataset. Bold and underscore denote the best and second-best performing methods respectively.

| Method | R→C | R→P | R→S | C→R | C→P | C→S | P→R | P→C | P→S | S→R | S→C | S→P | AVG |
|---|---|---|---|---|---|---|---|---|---|---|---|---|---|
| source [†] | 65.75 | 68.84 | 59.15 | 77.71 | 60.60 | 57.87 | 84.45 | 62.35 | 65.07 | 77.10 | 63.00 | 59.72 | 66.80 |
| BBSE (Lipton et al., 2018) [†] | 55.38 | 63.62 | 47.44 | 64.58 | 42.18 | 42.36 | 81.55 | 49.04 | 54.10 | 68.54 | 48.19 | 46.07 | 55.25 |
| PADA (Cao et al., 2018) [†] | 65.91 | 67.13 | 58.43 | 74.69 | 53.09 | 52.86 | 79.84 | 59.33 | 57.87 | 76.52 | 66.97 | 61.08 | 64.48 |
| MCD (Long et al., 2018) [†] | 61.97 | 69.33 | 56.26 | 79.78 | 56.61 | 53.66 | 83.38 | 58.31 | 60.98 | 81.74 | 56.27 | 66.78 | 65.42 |
| DAN (Long et al., 2015) [†] | 64.36 | 70.65 | 58.44 | 79.44 | 56.78 | 60.05 | 84.56 | 61.62 | 62.21 | 79.69 | 65.01 | 62.04 | 67.07 |
| F-DANN (Wu et al., 2019) [†] | 66.15 | 71.80 | 61.53 | 80.06 | 61.22 | 60.84 | 84.46 | 66.81 | 62.84 | 81.38 | 69.62 | 66.50 | 69.52 |
| UAN (You et al., 2019) [†] | 71.10 | 68.90 | 67.10 | 83.15 | 63.30 | 64.66 | 83.95 | 65.35 | 67.06 | 82.22 | 70.64 | 68.09 | 72.05 |
| JAN (Long et al., 2017) [†] | 65.57 | 73.58 | 67.61 | 85.02 | 64.96 | 67.17 | 87.06 | 67.92 | 66.10 | 84.54 | 72.77 | 67.51 | 72.48 |
| ETN (Cao et al., 2019) [†] | 69.22 | 72.14 | 63.63 | 86.54 | 65.33 | 63.34 | 85.04 | 65.69 | 68.78 | 84.93 | 72.17 | 68.99 | 73.99 |
| BSP (Chen et al., 2019) [†] | 67.29 | 73.47 | 69.31 | 86.50 | 67.52 | 70.90 | 86.83 | 70.33 | 68.75 | 84.34 | 72.40 | 71.47 | 74.09 |
| DANN (Ganin & Lempitsky, 2015) [†] | 63.37 | 73.56 | 72.63 | 86.47 | 65.73 | 70.58 | 86.94 | 73.19 | 70.15 | 85.73 | 75.16 | 70.04 | 74.46 |
| COAL (Tan et al., 2020) [†] | 73.58 | 75.37 | 70.50 | 89.63 | 69.98 | 71.29 | 89.81 | 68.01 | 70.49 | 87.97 | 73.21 | 70.53 | 75.89 |
| MDD+implicit | 75.54 | 74.30 | 70.02 | 88.17 | 70.50 | 70.30 | 87.94 | 72.03 | 72.29 | 88.85 | 76.12 | 71.21 | 76.44 |
| InstaPBM (Li et al., 2020) [†] | 80.10 | 75.87 | 70.84 | 89.67 | 70.21 | 72.76 | 89.60 | 74.41 | 72.19 | 87.00 | 79.66 | 71.75 | 77.84 |
| Sentry (Prabhu et al., 2021) [†] | **83.89** | 76.72 | 74.43 | **90.61** | **76.02** | **79.47** | **90.27** | **82.91** | 75.60 | **90.41** | **82.40** | 73.98 | **81.39** |
| Ours | 80.15 | **76.93** | **76.08** | 89.87 | 72.25 | 76.25 | 90.04 | 79.11 | **76.32** | 89.61 | 80.71 | **75.46** | 80.23 |

[†] Data of the baseline methods are cited from Prabhu et al. (2021)

(**Cl**), Product (**Pr**) and Art (**Ar**). In our experiments, we use the existing imbalanced Office-Home with **RS-UT** distributions generated in Tan et al. (2020) to train and test our approach. Since there are very limited samples in the art (**Ar**) domain, we only conduct domain adaptation tasks on the other three domains. **DomainNet** is a large-scale benchmark dataset for domain adaptation (Peng et al., 2019). Since there are mislabeled samples in some classes and domains, we follow Tan et al. (2020) and adopt only 40 common classes from four domains: Real (**R**), Clipart (**C**), Painting (**P**), and Sketch (**S**). Different from Office-31 and Office-Home, the selected samples in DomainNet already exhibit obvious label distribution shift in the source domain and target domain. So there is no need to sample this dataset again. Figure 5 illustrates the label distributions of imbalanced Office31 and imbalanced Office-Home datasets.

We use PyTorch to implement our approach. We train our model with the mini-batch SGD, a Nesterov momentum of 0.9, and a weight decay of 0.0005. The learning rate of classifiers is 10 times larger than that of feature extractor, and all the learning rates are adjusted by every iteration. In order to obtain the interpolated adversarial sample from a pair of source sample and target sample from the same class, we utilize a memory pool to store the pseudo labels of all the target data. The pseudo labels are updated in every iteration. Note that we utilize a class-balanced sampler on the source data, which can be referred to as N-way (number of classes per batch) and K-shot (number of examples per class). The coefficient $\alpha$ is set to 0.5 and $\beta$ is set to 0.05 for all the experiments. Following existing work on CDA, we adopt the per-class mean accuracy as our evaluation metric. All the experiments are implemented on Nvidia RTX A5000 platform.

Table 3: Accuracy of MDD and our full model on minority classes from imbalanced Office-Home (Rw→Pr). Bold denotes the best performing method.

| Method | Batteries | Bed | Bike | Bottle | Calculator | Chair | Clipboards | AVG |
|---|---|---|---|---|---|---|---|---|
| MDD (baseline) | 51.61 | 95.52 | 19.79 | 85.91 | 71.42 | 93.05 | 62.22 | 68.50 |
| Ours | **59.67** | **100.0** | **29.17** | **88.73** | **73.47** | **100.0** | **64.44** | **73.64** |

Table 4: Per-class average accuracy on Office-31 with RS→UT label shift. Bold and underscore denote the best and second-best performing methods respectively.

| Method | A→W | D→W | W→D | A→D | D→A | W→A | AVG |
|---|---|---|---|---|---|---|---|
| source | 71.77 | 90.86 | 93.06 | 72.25 | 59.03 | 58.34 | 74.21 |
| F-DANN (Wu et al., 2019) | 69.83 | 93.56 | 93.95 | 76.45 | 58.57 | 58.11 | 75.07 |
| COAL (Tan et al., 2020) | 81.18 | 91.12 | 95.46 | 81.67 | 66.08 | 66.60 | 80.35 |
| MDD+implicit (Jiang et al., 2020) | _85.79_ | **96.20** | **97.40** | _84.25_ | _68.11_ | _66.63_ | _83.06_ |
| Sentry (Prabhu et al., 2021) | 81.77 | 90.95 | 93.50 | 83.91 | 62.72 | 64.00 | 79.48 |
| Ours | **89.61** | _96.08_ | _97.08_ | **86.66** | **71.93** | **70.40** | **85.29** |

## 4.2 COMPARISON WITH STATE-OF-ART CDA METHODS

We compare our approach with several state-of-art methods for class imbalanced domain adaptation. The baseline methods can be divided into two categories. The first category of methods are specifically designed to solve the CDA problem, including Sentry (Prabhu et al., 2021), MDD+implicit Jiang et al. (2020), COAL (Tan et al., 2020), and F-DANN (Wu et al., 2019). The second category of methods aim to solve the standard unsupervised domain adaptation problem, including the InstaPBM (Li et al., 2020) and BSP (Chen et al., 2019).

Table 1 shows the per-class average accuracy and overall average accuracy of our approach and baselines on the imbalanced Office-Home dataset. The results of the baseline are cited from Prabhu et al. (2021). Our proposed model achieves the best results in 4 out of 6 tasks. The accuracies of our approach in these 4 tasks are more than 3% higher than the second-best results. In addition, our model obtains the second-best results in the Rw→CL and Pr→Cl tasks. Overall, our method achieves the highest average accuracy (i.e., 66.07%) among all the compared methods. We also investigate the class-specific accuracy of MDD (Zhang et al., 2019) and our method on minority classes, in order to understand how well our method addresses the data imbalance issue. Table 3 shows that our PAT method can significantly boost the performance on the minority classes from the imbalanced Office-Home dataset.

Table 2 shows the per-class average accuracies of our approach and baselines on the DomainNet dataset. Our approach achieves the best results in 4 tasks and second-best results in 8 tasks. In R→S, P→S and S→P, our model can achieve 76.08%, 75.46%, which are 1.65% and 1.48% higher than the second-best results, respectively. Though in some tasks our results are lower than Sentry (Prabhu et al., 2021), the overall average accuracy of our approach is comparable to that of Sentry.

We manually sample the standard Office-31 dataset and construct the imbalanced Office-31 dataset, in which the label distribution conforms to the Paredo distribution (Reed, 2001). We compare our model with current state-of-art methods that focus on the CDA problem. Among all the 6 tasks, our model achieves best results in 4 tasks. In these tasks, our results are 3.82%, 2.41%, 3.28% and 3.77% higher than the second best results. In the D→W and W→D tasks, the performance of our approach are only 0.12% and 0.32% less than the best results.

## 4.3 ABLATION STUDIES

We further investigate the performance of our approach from several aspects. First, we evaluate the contribution of each component in the loss function. Second, we evaluate the sensitivity of the model to the change of the hyper-parameters. Third, we evaluate the performance of our method on the standard unsupervised domain adaptation problem. We have also evaluated: (1) the effect of the number of the iteration in the inner maximization of PAT; and (2) the integration of PAT and other domain adaptation methods including CDAN, CDANE and Sentry, and reported the results in the Appendix due to space limit.

Table 5: Per-class average accuracy of MDD, Ours w/o IAS, Ours w/o CA, and our full model on three tasks from imbalanced Office-Home dataset. Bold denotes the best performing method.

| Method | Rw→Pr | Rw→Cl | Pr→Rw | Pr→Cl | Cl→Rw | Cl→Pr | AVG |
|---|---|---|---|---|---|---|---|
| MDD (baseline) [†] | 75.96 | 47.38 | 71.56 | 42.73 | 57.46 | 58.76 | 58.98 |
| Ours w/o IAS [†] | 78.38 | 51.85 | 75.72 | 48.31 | 67.16 | 65.80 | 64.26 |
| Ours w/o CA [†] | 77.37 | 53.02 | 76.12 | 47.08 | 64.99 | 64.73 | 63.61 |
| Ours [†] | **79.88** | **54.68** | **77.32** | **50.21** | **67.29** | **67.04** | **66.07** |

[†] We adopt class-balanced source sampling on all these methods.

Table 6: Accuracy of MDD and our model on four tasks from standard Office-Home. Bold denotes the best performing method.

| Method | Rw→Pr | Ar→Rw | Cl→Pr | Pr→Cl | AVG |
|---|---|---|---|---|---|
| MDD (baseline) [†] | 82.3 | 77.8 | 71.4 | 53.6 | 71.3 |
| Ours | **84.7** | **80.2** | **75.1** | **54.3** | **73.6** |

[†] Data of the baseline methods are cited from Zhang et al. (2019) .

The proposed pairwise adversarial training approach includes two major components, i.e., interpolated adversarial samples (IAS) and centroid alignment (CA). We choose three tasks from each of the three datasets separately. Table 5 shows the performance of our backbone model MDD, Ours w/o IAS, Ours w/o CA, and our full model on the imbalanced Office-Home dataset. Detailed results on the other two datasets are provided in the Appendix. Ours w/o IAS and Ours w/o CA consistently achieve better performance than the baseline model MDD, which validates the effectiveness of both methods in dealing with the biased label distribution shift. Our full model further improves the classification accuracy, demonstrating the complementary roles of pairwise adversarial samples and centroid alignment in our approach.

Furthermore, we evaluate the sensitivity of our model to the change of two hyperparameters $\alpha$ and $\beta$ . In particular, we first set $\beta$ to 0.05 and choose $\alpha$ from $[0, 2.5]$. Then, $\alpha$ is set to 0.5, and $\beta$ is chosen from $[0, 0.25]$. Figure 6 shows the per-class average accuracy of our model when varying the hyper-parameter values. It shows that our model is not very sensitive to the settings of hyper-parameters in a relatively wide range.

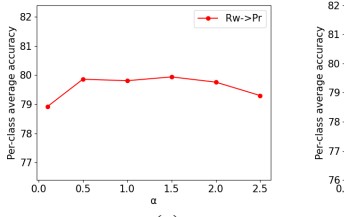 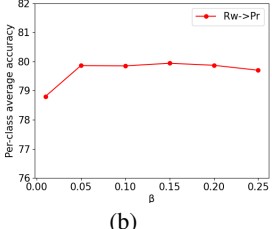

Figure 6: Average accuracy of our model with: (a) varying $\alpha$ when $\beta = 0.05$, and (b) varying $\beta$ when $\alpha = 0.5$ on Rw→Pr in imbalanced Office-Home.

Finally, we evaluate our method in the standard UDA setting, by comparing the results against the baseline MDD (Zhang et al., 2019) on the standard Office-Home dataset. The results from Table 6 shows that the classification accuracy in all the four tasks are improved after incorporating our interpolation method, which proves the effectiveness of our method in unsupervised domain adaptation problem.

## 5 CONCLUSION

In this paper, we propose a pairwise adversarial training approach to tackle the class-imbalanced unsupervised domain adaptation (CDA) problem. Our approach generates interpolated adversarial samples across source and target domains. In order to alleviate the biased label distribution shift issue, we use the interpolated adversarial samples to augment the training data (especially the minority classes) and meanwhile adopt the centroid alignment strategy to explicitly align source and target domains. Experimental results on three CDA benchmark datasets show that, our approach yields competing performance compared with state-of-art CDA methods.

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

## A APPENDIX

We evaluate the effect of the number of iterations of the inner maximization in pairwise adversarial training. We choose Rw→Pr from Office-Home, R→C from DomainNet and A→W from Office-31 as our evaluation tasks. The results are illustrated in Figure 7. It shows that the performance of our approach will generally increase with more iterations. At some points the performance experiences the slight fluctuation. Though performance reaches the maximum value at iteration of 12, it is relatively time-consuming in the training process.

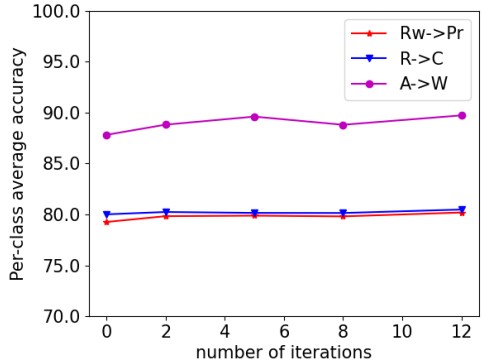

Figure 7: Per-class average accuracy of our model with different number of iterations.

Table 7 and Table 8 show the per-class average accuracy of MDD, Ours w/o IAS, Ours w/o CA, and our full model on three tasks from the DomainNet dataset and Office-31 dataset, respectively. The results demonstrate the effectiveness of two major components (i.e., interpolated adversarial samples and centroid alignment) in our approach.

Table 7: Per-class average accuracy of MDD, Ours w/o IAS, Ours w/o CA, and our full model on three tasks from DomainNet dataset. Bold denotes the best performing method.

| Method | R→C | C→P | C→R | AVG |
|---|---|---|---|---|
| MDD (baseline) [†] | 75.30 | 70.38 | 87.94 | 77.87 |
| Ours w/o IAS [†] | 77.03 | 69.43 | 88.60 | 78.35 |
| Ours w/o CA [†] | 77.65 | 71.25 | 88.49 | 79.13 |
| Ours [†] | **80.15** | **72.25** | **89.87** | **80.76** |

[†] We adopt class-balanced source sampling on all these methods.

Table 8: Per-class average accuracy of MDD, Ours w/o IAS, Ours w/o CA, and our full model on imbalanced Office-31 dataset. Bold denotes the best performing method.

| Method | A→W | W→D | W→A | AVG |
|---|---|---|---|---|
| MDD (baseline) [†] | 85.86 | 96.12 | 65.20 | 82.39 |
| Ours w/o IAS [†] | 87.41 | 96.44 | 68.42 | 84.09 |
| Ours w/o CA [†] | **90.58** | 96.55 | 69.99 | 85.71 |
| Ours [†] | 89.61 | **97.08** | **70.04** | **85.58** |

[†] We adopt class-balanced source sampling on all these methods.

Furthermore, as our PAT method is an independent module, which could be combined with other domain alignment techniques such as CDAN and Sentry. We integrate PAT with CDAN and CDANE, and report the experimental results on the RS-UT Office-Home dataset in Table 9. Results show that our PAT module could significantly improve the performance: 54.07%→60.38% for CDAN, and

55.70%→62.61% for CDANE. The results in Table 10 further validate that our PAT method could boost the performance of the state-of-the-art CDA method Sentry.

Table 9: Per-class average accuracy of CDAE, CDANE and our methods on imbalanced Office-Home dataset with RS→UT label shift. Bold denotes the best performing method.

| Model | Rw→Pr | Pr→Cl | Cl→Pr | AVG |
|---|---|---|---|---|
| CADN | 70.78 | 38.46 | 52.99 | 54.07 |
| CDAN+PAT (ours) | 77.44 | 42.16 | 61.54 | 60.38 |
| CDANE | 72.16 | 40.39 | 54.57 | 55.70 |
| CDANE+PAT (ours) | **78.59** | **44.20** | **65.04** | **62.61** |

Table 10: Per-class average accuracy of Sentry and our method on DomainNet dataset. Bold denotes the best performing method.

| Model | R→C | P→R | S→R | AVG |
|---|---|---|---|---|
| Sentry | 83.89 | 90.27 | 90.41 | 88.19 |
| Sentry + PAT (ours) | **86.42** | **91.33** | **91.37** | **89.71** |

