# OpenReview forum: "Pairwise Adversarial Training for Unsupervised Class-imbalanced Domain Adaptation"
_ICLR.cc/2022/Conference — ICLR 2022 Submitted_

### Official Review · Reviewer_sUHK · 2021-11-01

**Correctness:** 3
**Technical Novelty And Significance:** 3
**Empirical Novelty And Significance:** 2
**Recommendation:** 5
**Confidence:** 4

**Main Review:**

This paper solve the class-imbalanced problem from two aspects:
1. pairwise adversarial training and generate the samples on the interpolated line from a source sample to a target sample of the same class. And samples from the minority class will have larger chance.
2. Align the conditional feature distributions of source and target domains by explicitly matching the centroids of two domains.

However, my major concerns are as follows:
1. During the data generating process, the pseudo-labels of the target data are still not accurate. Won't there still exist error accumulation?
2. The improvements of the experimental results are not obvious.

**Summary Of The Paper:**

This paper proposes an adversarial data augmented method to solve the class-imbalanced problem in domain adaptation.

**Summary Of The Review:**

The novelty is limited and some statements should be explained further so that the advantage of the proposed method could be more obvious.

---

> ### Author Response · Authors · 2021-11-22
> **Reply to Reviewer sUHK**
>
> We thank the reviewer for providing constructive comments. In the following we provide detailed responses to these questions.
>
> **Q1. During the data generating process, the pseudo-labels of the target data are still not accurate. Won't there still exist error accumulation?**
>
> R1. We agree with the reviewer that the quality of pseudo labels is essential in our approach. In fact, our approach deals with the potential issue of error accumulation due to incorrect pseudo labeling from two aspects.
>
> First, in our training, we only choose pseudo labels of the target samples with high confidence (i.e., p > 0.8). To understand the quality of target pseudo labels, we compared them with ground truth target labels and observed an accuracy around 88%. Thus, most of the pseudo labels used in our approach are very reliable.
>
> Second, even if the pseudo label is incorrect, there is still a chance that the interpolated adversarial sample (IAS) will be generated within the boundary as expected. Specifically, the misclassified target samples often appear near the decision boundary. So, even if the target sample is from a different class, the adversarial sample generated from the pair of source sample and target sample may not violate the decision boundary. Moreover, the adversarial samples in our approach are generated dynamically, and the adverse effect of bad adversarial samples could be mitigated.
>
> We have included the above discussions in our revised paper.
>
> **Q2. The improvements of the experimental results are not obvious**
>
> R2. In the following we’d like to clarify the significance of our method and results from three aspects.
>
> First, our PAT method is an independent module which is primarily designed for the CDA problem. PAT can be integrated with other UDA models to deal with the data imbalance issue. We integrate PAT with a representative UDA method, MDD, and significantly improve the performance (in terms of per-class average accuracy) of MDD in class-imbalanced settings. On the Office-Home dataset, the average accuracy is improved by **over 10%**, from 55.44% (MDD) to 66.07% (Ours, i.e., MDD+PAT). On the DomainNet dataset, the average accuracy is improved from 76.44% (MDD) to 80.23% (Ours, i.e., MDD+PAT). We have also integrated PAT with other two UDA methods (i.e., CDAN and CDANE), and results show that PAT improves the average accuracy by about **7%**.
>
> Second, our method (i.e., MDD+PAT) outperforms the state-of-the-art CDA method Sentry in 14 out of 24 tasks on three datasets. Moreover, our PAT method is an independent module that could be integrated with existing UDA or CDA methods. Our additional results below show that PAT could further boost the performance of Sentry on the DomainNet dataset.
>
> | Model/Task | R->C | P->R | S->R | Ave |
> |:---------|:-------:|:-------:|:-------:|:-------:|
> |Sentry | 83.89 | 90.27 | 90.41 | 88.19 |
> |Sentry+PAT (ours) | **86.42** | **91.33**  | **91.37** | **89.71** |
>
> Third, apart from the experiments on CDA problems, we also conduct experiments on traditional UDA problems using our method. Experimental results on the OfficeHome dataset show that our method consistently outperforms the UDA baselines in all the tasks.
>
> |  Model | Rw->Pr | Ar->Rw | Cl->Pr | Pr->Cl |   Ave |
> |:---- |:------:| :-----:| :-----:| :-----:| :----:|
> | GSDA [1]  |  83.1    |  79.4  |   73.3 | 53.2   | 72.3  |
> | GVB-GD [2]  | 84.3    |  79.8  |   74.1 | **55.1**  | 73.3 |
> |MDD  | 82.3   | 77.8   | 71.4   | 53.6   | 71.3  |
> |MDD + PAT (ours) | **84.7**  | **80.2**  | **75.1**  | 54.3  | **73.6**  |
>
> [1] Hu, Lanqing, et al. "Unsupervised domain adaptation with hierarchical gradient synchronization." Proceedings of the IEEE/CVF Conference on Computer Vision and Pattern Recognition. 2020.
>
> [2] Cui, Shuhao, et al. "Gradually vanishing bridge for adversarial domain adaptation." Proceedings of the IEEE/CVF Conference on Computer Vision and Pattern Recognition. 2020.

---

> > ### Author Response · Authors · 2021-11-29
> > **Response to Reviewer sUHK**
> >
> > Thank you again for reviewing our paper. We have answered all the questions and provided additional experimental results. If there is anything unclear, we will address it further.

---

### Official Review · Reviewer_Cnqr · 2021-11-02

**Correctness:** 4
**Technical Novelty And Significance:** 3
**Empirical Novelty And Significance:** 3
**Recommendation:** 5
**Confidence:** 5

**Main Review:**

Strength:
1. the studied problem called class-imbalanced domain adaptation is interesting and practical
2. the proposed method especially the IAS part is simple and easy to understand
3. the results on CDA are impressive

Weakness:
1. the part called semantic centroid alignment (CA) within the proposed PAT is commonly used in the domain adaptation field (Xie et al., 2018), and the moving average strategy and the balancing strategy are also not new

2. the authors aim to address the CDA problem, however, it sounds that PAT could work well for vaninna UDA problems, the authors are suggested to perform comparison with UDA methods on these datasets. For example, the subsets of the VISDA-2017 dataset are not label-balanced, PAT could be compared with recent UDA methods [a-b] directly w.r.t. the per-class accuracy.

[a]. Liang, Jian, Dapeng Hu, and Jiashi Feng. "Do we really need to access the source data? source hypothesis transfer for unsupervised domain adaptation." International Conference on Machine Learning. PMLR, 2020.

[b]. Na, Jaemin, et al. "FixBi: Bridging Domain Spaces for Unsupervised Domain Adaptation." Proceedings of the IEEE/CVF Conference on Computer Vision and Pattern Recognition. 2021.

3. besides, it seems that PAT is independent on MDD (adopted in this paper), could the authors further show its combination with else domain alignment techniques like CDAN? And the results in Table 4 only show 3 out of 6 tasks on the OfficeHome dataset, and the contribution of IAS seems not significant (I think IAS is main contribution of this paper).

4. minor concerns: how to determine the parameters like \alpha and \beta within PAT, results in Fig.7 show PAT is somewhat sensitive to these parameters.

**Summary Of The Paper:**

This paper proposes a new method called Pairwise Adversarial Training (PAT) that augments training data for class-imbalanced domain adaptation (CDA). Different from vanilla unsupervised domain adaptation, the label distributions of different distributions are quite different in CDA. The proposed PAT approach mainly consists of two part, centroid alignment (CA) and interpolated adversarial samples (IAS). Experiments on several benchmarks verify the effectiveness of PAT for the CDA problem.

**Summary Of The Review:**

Class-imbalanced domain adaptation is an interesting and important problem in the UDA field. The strategy of interpolated adversarial samples (IAS) is new and interesting, however, the overall novelty of the proposed method sounds not much high for a top-tier conference. In addition, the experiments do not fully verify the effectiveness of PAT especially for IAS, thus I vote for "weak reject."

---

> ### Author Response · Authors · 2021-11-22
> **Reply to Reviewer Cnqr**
>
> We thank the reviewer for providing constructive comments. In the following we provide detailed responses to these questions.
>
> **Q1. The authors aim to address the CDA problem, however, it sounds that PAT could work well for vanilla UDA problems.**
>
> R1. Yes. Our PAT method also works well for UDA problems. The table below shows the accuracy of several models on standard OfficeHome.
>
> We conducted additional experiments to evaluate the performance of our PAT approach on the standard UDA problem. The table below shows the accuracy of MDD (Baseline) and MDD+PAT (Ours) on the OfficeHome dataset. Results show that the classification accuracy in all the four tasks are improved after incorporating  our interpolation method. We have included the new results in the revised paper.
>
> | Model/Task  | Rw->Cl  |  Ar->Rw  |  Cl->Pr  | Pr->Cl | Average  |
> |:-------------|:----------:|:-----------:|:------------:|:---------:|:------------:|
> | MDD |  82.3  | 77.8  |  71.4  |  53.6  |  71.3   |
> | MDD+PAT (ours) | **84.7**  | **80.2** | **75.1** | **54,3** | **73.6** |
>
> Besides, we also add the PAT methods on the suggested SHOT method [a]. The table below shows the accuracy in four tasks on the standard OfficeHome benchmark. And there is slight improvement on the three out of four tasks. The improvement is not significant on the standard UDA problem, since our PAT method is designed to address the class-imbalanced domain adaptation problem.
>
> | Model/Task  | Rw->Cl  |  Ar->Rw  |  Cl->Pr  | Pr->Cl | Average  |
> |:-------------|:----------:|:-----------:|:------------:|:---------:|:------------:|
> | SHOT [a] |  84.3  | 81.5  |  78.2  |  54.9  |  74.7   |
> | SHOT+PAT (ours) | 84.5  | 80.7 | 78.3 | 55.1 | 74.7 |
>
> [a] Liang, Jian, Dapeng Hu, and Jiashi Feng. "Do we really need to access the source data? source hypothesis transfer for unsupervised domain adaptation." International Conference on Machine Learning. PMLR, 2020.
>
> **Q2. Besides, it seems that PAT is independent of MDD (adopted in this paper), could the authors further show its combination with other domain alignment techniques like CDAN? And the results in Table 4 only show 3 out of 6 tasks on the OfficeHome dataset**
>
> R2. Thanks for the valuable suggestion. Yes. PAT could be combined with other domain alignment techniques such as CDAN. We integrate PAT with CDAN and CDANE, and report the experimental results on the RS-UT OfficeHome dataset in the following table. Results show that our PAT module could significantly improve the performance: 54.07%->60.38% for CDAN, and 55.70%->62.61% for CDANE.
>
> | Model/Task | Rw->Pr | Pr->Cl | Cl->Pr | Ave |
> |:---------|:----------:|:---------:|:--------:|:-----:|
> | CADN | 70.78 | 38.46 | 52.99 | 54.07 |
> | CDAN+PAT (ours) | 77.44 | 42.16 | 61.54 | 60.38 |
> | CDANE | 72.16 | 40.39 | 54.57 | 55.70 |
> | CDANE+PAT (ours) | **78.59** | **44.20** | **65.04** | **62.61** |
>
> We have expanded Table 4 in our paper by including results on all the 6 tasks. Results demonstrate the contribution of IAS in every task.
>
> | Model/Task  | Rw-Pr  |  Rw->Cl  |  Pr->Rw  | Pr->Cl |  Cl->Rw  |  Cl->Pr  | Average  |
> |:---------|:----------:|:-----------:|:------------:|:---------:|:------------:|:---------:|:------------:|
> | MDD | 75.96 | 47.38 | 71.56 | 42.73 | 57.48 | 58.76 | 58.98 |
> | Ours w/o IAS | 78.38 | 51.85 | 75.72 | 48.31 | 67.16 | 64.16 | 64.26 |
> | Ours w/o CA | 77.37 | 53.02 | 76.12 | 47.08 | 64.99 | 63.06 | 63.61 |
> | MDD+PAT(ours) | **79.88** | **54.68** | **77.32** | **50.21** | **67.29** | **67.04** | **66.07** |
>
>
> **Q3. minor concerns: how to determine the parameters like \alpha and \beta within PAT, results in Fig.7 show PAT is somewhat sensitive to these parameters.**
>
> R4. In Fig. 7, the range of y-axis is from 77% to 82%, and the variation of accuracy is mostly within 1%. So, the performance of PAT is actually not very sensitive to the setting of these hyperparameters. In the experiments we empirically set these parameters, following the commonly used evaluation protocol in the domain adaptation literature.

---

> > ### Author Response · Authors · 2021-11-29
> > **Response to Reviewer Cnqr**
> >
> > Thank you again for reviewing our paper. We have answered all the questions and provided additional experimental results. If there is anything unclear, we will address it further.

---

### Official Review · Reviewer_p4yb · 2021-11-02

**Correctness:** 3
**Technical Novelty And Significance:** 3
**Empirical Novelty And Significance:** Not applicable
**Recommendation:** 5
**Confidence:** 5

**Main Review:**

Strength:

1. The interpolation based data augmentation method is interesting.
2. The overall approach is simple and easy to implement.

Weakness:
1. Novelty is limited. The interpolation is similar to mix-up and follow-up works. Relation with mix-up should be discussed here.
2. The proposed method does not outperform the baseline Sentry.
3. Some statements are based on intuition but not well supported. For instance, the proposed method is claimed to help the model generalize better on the target domain for minority classes but the corresponding experiments are lacking.

**Summary Of The Paper:**

This work proposes a method for solving the UDA problem with imbalanced class, which is a sub-problem of UDA. The challenge lies in how to handle the difficulties introduced by imbalanced classes. To this end, this work proposes a new data augmentation strategy, that is taking the interpolation of two samples from the same class but from different domains as the augmented samples. The traditional MMD loss and a class centroid distance based loss are also imposed for the model training. Experiments on multiple benchmark datasets are conducted.

**Summary Of The Review:**

Overall, this work introduces a neat idea, that is taking the interpolation of two anchor samples to augment the minority class. But overall the paper is lacking in depth and the experiment results are not very convincing.

---

> ### Author Response · Authors · 2021-11-22
> **Reply to Reviewer p4yb**
>
> We thank the reviewer for providing constructive comments. In the following we provide detailed responses to these questions.
>
> **Q1. The interpolation is similar to mix-up and follow-up works. Relation with mix-up should be discussed here.**
>
> R1. Although both mix-up (and its follow-up works) and our method use interpolation to generate new samples, they have significant differences, as discussed below.
>
> First, mix-up and our method focus on different tasks. Mix-up leverages interpolation to address the classification problem in a single domain, while our method focuses on the domain adaptation problem that involves a source domain and a target domain. Although a few follow-up work of mix-up try to address the domain adaptation problem, they are different from our method as we focus on the data imbalance problem in domain adaptation.
>
> Second, the ideas for generating samples and labels in mix-up and our method are different. Mix-up creates virtual samples from two randomly chosen samples, and the labels of the virtual samples are also interpolations of these two samples. While in our method, an adversarial sample is generated using a pair of samples from two domains with the same label. The generated adversarial sample is expected to have the same semantic meaning as the original pair of samples.
>
> Third, the strength of interpolation in mix-up and our method is controlled in different ways. Mix-up needs to manually define a hyper-parameter to control the strength of interpolation. However, in our method, the parameter for controlling the strength of interpolation is adaptively updated along with the adversarial training.
>
> We have included the above discussions in our revised paper.
>
> **Q2. The proposed method does not outperform the baseline Sentry.**
>
> R2. Sentry (ICCV’21) is a major baseline in our experiments. In the following we discuss the differences between our approach and Sentry in terms of the experimental results and methodology.
>
> First, our approach outperforms Sentry in terms of the average accuracy on the Office-Home dataset and Office-31 dataset. In detail, our method outperforms Sentry in 4 out of 6 tasks on Office-Home dataset, and all the 6 tasks on the Office-31 dataset. On the DomainNet dataset, our approach obtains the best results in 4 tasks, and obtains similar results than Sentry in the other tasks. In summary, our approach outperforms Sentry in most tasks (i.e., 14 out of 24 tasks) on the Office-Home, Office-31, and DomainNet datasets.
>
> Second, our PAT method and Sentry address the class-imbalance domain adaptation problem from different perspectives. Our method aims to resolve the class imbalanced problem by augmenting training data, while Sentry adopts mutual information maximization to assist domain alignment in the presence of label-distribution shift. In fact, our method and Sentry are complementary to each other, and their integration will lead to further performance gain. The table below shows that our PAT method could further boost the performance of Sentry on the DomainNet dataset. Full results will be included in the final version.
>
> | Model/Task | R->C | P->R | S->R | Ave |
> |:---------|:-------:|:-------:|:-------:|:-------:|
> |Sentry | 83.89 | 90.27 | 90.41 | 88.19 |
> |Sentry+PAT (ours) | **86.42** | **91.33**  | **91.37** | **89.71** |
>
> **Q3. The proposed method is claimed to help the model generalize better on the target domain for minority classes, but the corresponding experiments are lacking.**
>
> R3. Thanks for the suggestion. We analyzed the performance of our PAT method on some minority classes and summarized the results in the table below. Results show that PAT significantly boosts the performance of the minority classes. These results have been included in our revised paper.
>
> | Model/Class | Batteries | Bed | Bike | Bottle | Calculator | Chair | Clipboards | Ave |
> | :---------| :----------:|:-----:|:------:|:--------:|:-------------:|:--------:|:--------------:|:-----:|
> | MDD | 51.61 | 95.52 | 19.79 | 85.91 | 71.42 | 93.05 | 62.22 | 68.50 |
> | MDD+PAT (ours)| **59.67** | **100.0** | **29.17** | **88.73** | **73.47** | **100.0** | **64.44** | **73.64** |

---

> > ### Author Response · Authors · 2021-11-29
> > **Response to Reviewer p4yb**
> >
> > Thank you again for reviewing our paper. We have answered all the questions and provided additional experimental results. If there is anything unclear, we will address it further.

---

### Official Review · Reviewer_orbW · 2021-11-03

**Correctness:** 3
**Technical Novelty And Significance:** 2
**Empirical Novelty And Significance:** 2
**Recommendation:** 5
**Confidence:** 5

**Main Review:**


- Since the target domain is unlabeled, the interpolation between a source sample and a target sample of the same class is not reliable, which may face the same error accumulation issue as the pseudo-labeling methods. If the pseudo-label is not correct, the augmented data may not only cause the error accumulation but also damage the domain alignment.

- How does the interpolated adversarial sample generation method deal with the data imbalance issue in the target domain? It seems that the generation process that generates data merely resorts to the guidance of the class probability of the source domain data.

- The interpolation method looks quite general, will this method also work well on the UDA problem?

- In experiments, more discussions are required for explaining why the proposed method performs worse than Sentry (Prabhu et al., 2021).


**Summary Of The Paper:**

This paper proposes a pairwise adversarial training approach for class-imbalanced domain adaptation. Specifically, the adversarial samples are generated from the interpolated line of the aligned pairwise source domain samples and target domain samples. The generated adversarial data can augment the training data and help enhance the robustness of models.

**Summary Of The Review:**

In general, the proposed method is somewhat novel. However, the method is not very well justified and some parts of the method are unclear.

---

> ### Author Response · Authors · 2021-11-22
> **Reply to Reviewer orbW (Part 1)**
>
> We thank the reviewer for providing constructive comments. In the following we provide detailed responses to these questions.
>
> **Q1. If the pseudo-label is not correct, the augmented data may not only cause the error accumulation but also damage the domain alignment.**
>
> R1. We agree with the reviewer that the quality of pseudo labels is essential in our approach. In fact, our approach deals with the potential issue of incorrect pseudo labeling from two aspects.
>
> First, in our training, we only choose pseudos labels of the target samples with high confidence (i.e., p > 0.8). To understand the quality of target pseudo labels, we compared them with ground truth target labels and observed an accuracy around 88%. Thus, most of the pseudo labels used in our approach are very reliable.
>
> Second, even if the pseudo label is incorrect, there is still a chance that the interpolated adversarial sample (IAS) will be generated within the boundary as expected. Specifically, the misclassified target samples often appear near the decision boundary. So, even if the target sample is from a different class, the adversarial sample generated from the pair of source sample and target sample may not violate the decision boundary. Moreover, the adversarial samples in our approach are generated dynamically, and the adverse effect of bad adversarial samples could be mitigated.
>
> We have included the above discussions in our revised paper.
>
> **Q2. How does the interpolated adversarial sample generation method deal with the data imbalance issue in the target domain?**
>
> R2. In unsupervised domain adaptation, the training of the classifier heavily relies on the source domain that is fully labeled. Previous studies and our investigation found that, when the source domain is imbalanced, the model performance on target domains will be significantly dropped, especially when the target domain is also imbalanced. Given this intuition, our idea is to explicitly address the data imbalance issue in the source domain, and train an unbiased model by exploiting interpolated adversarial samples and aligning source and target domains. As a result, the generalization ability of the model will be improved and the data imbalance issue in the target domain could be implicitly addressed. In our approach, the minority classes in the source domain have a larger chance to generate adversarial samples, as described in Section 3.2.2 (Page 5). We then adapt the model to the target domain through domain adversarial learning and centroid alignment.
>
> Moreover, to illustrate the performance of PAT on dealing with the data imbalance issue in the target domain, the table below shows the accuracy of MDD and MDD+ PAT on the minority classes from the RS-UT OfficeHome dataset. Results show that PAT significantly boosts the performance of the minority classes.
>
> | Model/Class | Batteries | Bed | Bike | Bottle | Calculator | Chair | Clipboards | Ave |
> | :---------| :----------:|:-----:|:------:|:--------:|:-------------:|:--------:|:--------------:|:-----:|
> | MDD | 51.61 | 95.52 | 19.79 | 85.91 | 71.42 | 93.05 | 62.22 | 68.50 |
> | MDD+PAT (ours)| **59.67** | **100.0** | **29.17** | **88.73** | **73.47** | **100.0** | **64.44** | **73.64** |

---

> ### Author Response · Authors · 2021-11-22
> **Reply to Reviewer orbW (Part 2)**
>
> **Q3. The interpolation method looks quite general, will this method also work well on the UDA problem?**
>
> R3. Yes. Our interpolation method also works well on the UDA problem. We conducted additional experiments to evaluate the performance of our PAT approach on the standard UDA problem. The table below shows the accuracy of MDD (Baseline) and MDD+PAT (Ours) on the OfficeHome dataset. Results show that the classification accuracy in all the four tasks are improved after incorporating  our interpolation method. We have included the new results in the revised paper.
>
> | Model/Task  | Rw->Cl  |  Ar->Rw  |  Cl->Pr  | Pr->Cl | Average  |
> |:-------------|:----------:|:-----------:|:------------:|:---------:|:------------:|
> | MDD |  82.3  | 77.8  |  71.4  |  53.6  |  71.3   |
> | MDD+PAT (ours) | **84.7**  | **80.2** | **75.1** | **54,3** | **73.6** |
>
> **Q4. In experiments, more discussions are required for explaining why the proposed method performs worse than Sentry.**
>
> R4. Thanks for the suggestion. Sentry (ICCV’21) is a major baseline in our experiments. In the following we discuss the differences between our approach and Sentry in terms of the experimental results and methodology.
>
> First, our approach outperforms Sentry in terms of the average accuracy on the Office-Home dataset and Office-31 dataset. In detail, our method outperforms Sentry in 4 out of 6 tasks on Office-Home dataset, and all the 6 tasks on the Office-31 dataset. On the DomainNet dataset, our approach obtains the best results in 4 tasks, and obtains similar results than Sentry in the other tasks. In summary, our approach outperforms Sentry in most tasks (i.e., 14 out of 24 tasks) on the Office-Home, Office-31, and DomainNet datasets.
>
> Second, our PAT method and Sentry address the class-imbalance domain adaptation problem from different perspectives. Our method aims to resolve the class imbalanced problem by augmenting training data, while Sentry adopts mutual information maximization to assist domain alignment in the presence of label-distribution shift. In fact, our method and Sentry are complementary to each other, and their integration will lead to further performance gain. The table below shows that our PAT method could further boost the performance of Sentry on the DomainNet dataset. Full results will be included in the final version.
>
> | Model/Task | R->C | P->R | S->R | Ave |
> |:---------|:-------:|:-------:|:-------:|:-------:|
> |Sentry | 83.89 | 90.27 | 90.41 | 88.19 |
> |Sentry+PAT (ours) | **86.42** | **91.33**  | **91.37** | **89.71** |

---

> > ### Author Response · Authors · 2021-11-29
> > **Response to Reviewer orbW**
> >
> > Thank you again for reviewing our paper. We have answered all the questions and provided additional experimental results. If there is anything unclear, we will address it further.

---

### Decision · Program_Chairs · 2022-01-20

**Decision:**

Reject

**Comment:**

This paper aims to address the imbalanced class problem in unsupervised domain adaptation. The challenge lies in how to handle the difficulties introduced by imbalanced classes. To this end, this work proposes a new data augmentation strategy by taking the interpolation of two samples from the same class but from different domains as the augmented samples. The experiments demonstrate promising performance on the class-imbalanced domain adaptation datasets.

However, there are several concerns raised by the reviewers. 1) The interpolation between a source and target sample of the same class can potentially be unreliable as the pseudo label methods. 2) Some statements are based on intuition but not well supported by either theoretical analysis or experimental evaluations. 3) The proposed method is inferior to baseline methods on some datasets, it would be helpful to have further analysis of the advantages and limitations of the proposed method.

Overall, the paper provides some new and interesting ideas. However, given the above concerns, the novelty and significance of the paper will degenerate. More discussions on the principles behind the proposed method and more experimental studies are needed. Addressing the concerns needs a significant amount of work. Although we think the paper is not ready for ICLR in this round, we believe that the paper would be a strong one if the concerns can be well addressed.